# ROBUST DIFFUSION GAN USING SEMI-UNBALANCED OPTIMAL TRANSPORT

## ABSTRACT

Diffusion models, a type of generative model, have demonstrated great potential for synthesizing highly detailed images. By integrating with GAN, advanced diffusion models like DDGAN (Xiao et al., 2022) could approach real-time performance for expansive practical applications. While DDGAN has effectively addressed the challenges of generative modeling, namely producing high-quality samples, covering different data modes, and achieving faster sampling, it remains susceptible to performance drops caused by datasets that are corrupted with outlier samples. This work introduces a robust training technique based on semi-unbalanced optimal transport to mitigate the impact of outliers effectively. Through comprehensive evaluations, we demonstrate that our robust diffusion GAN (RDGAN) outperforms vanilla DDGAN in terms of the aforementioned generative modeling criteria, i.e., image quality, mode coverage of distribution, and inference speed, and exhibits improved robustness when dealing with both clean and corrupted datasets.

## 1 INTRODUCTION

Despite their relatively recent introduction, diffusion models have experienced rapid growth and garnered significant attention in the research community. These models effectively reverse the diffusion process from Gaussian random noise inputs into clean, high-quality images. The models have found utility across diverse data domains, with their most remarkable successes being in the realm of image generation. Notably, diffusion models outperform state-of-the-art generative adversarial networks (GANs) in terms of the quality of generated content on various datasets, as shown in (Dhariwal & Nichol, 2021; Saharia et al., 2022). Furthermore, they exhibit superior mode coverage, as discovered by (Song et al., 2021b; Huang et al., 2021; Kingma et al., 2021), and offer adaptability in handling a wide range of conditional inputs, including semantic maps, text, representations, and images, as highlighted in the work of (Rombach et al., 2022; Meng et al., 2021; Wang et al., 2022). This flexibility has led to their application in various areas, such as text-to-image generation, image-to-image translation, image inpainting, image restoration, and more. Recent advancements in text-to-image generative models, based on diffusion techniques as proposed by (Ramesh et al., 2022; Saharia et al., 2022), have enabled the generation of highly realistic images from textual inputs. Furthermore, personalized text-to-image diffusion models, such as (Ruiz et al., 2022; Le et al., 2023), have found extensive applications in various real-world scenarios.

Nonetheless, despite their immense potential, diffusion models are hindered by a significant weakness: their slow computational speed. This limitation prevents their widespread adoption, contrasting them with GANs. The foundational Denoising Diffusion Probabilistic Models (DDPMs) by (Ho et al., 2020) require a thousand sampling steps to attain the desired output quality, resulting in minutes of computation for a single image generation. Although several techniques have been devised to reduce inference time (Song et al., 2021a; Lu et al., 2022; Zhang & Chen, 2022), primarily through the reduction of sampling steps, they still take seconds to generate a $32 \times 32$ image—roughly 100 times slower than GANs. Diffusion GAN (DDGAN), as introduced by (Xiao et al., 2022), has effectively tackled the challenge of modeling complex multimodal distributions, particularly when dealing with large step sizes, through the utilization of generative adversarial networks. This innovative approach has led to a significant reduction in the number of denoising steps required, typically just a few (e.g., 2 or 4). Consequently, inference times have been drastically reduced to mere fractions of a second.

In practice, datasets collected unintentionally inevitably contain outliers during the data collection process. These outliers can significantly harm the performance of generative models. However, because of the dataset's large scale, removing these outlier samples can be a daunting and time-consuming task. As a result, there is a strong demand for developing robust generative models that can counteract the negative effects of noisy datasets. While DDGAN has succeeded in striking a good balance among three crucial aspects of generative models—mode coverage, high-resolution synthesis, and fast sampling—we have observed a notable decline in DDGAN's performance when it encounters noisy data containing outliers. This limitation hinders its practical application. Given that DDGAN comprises multiple GAN models between two consecutive timesteps, we delve into the evolution of GANs to find a solution to the noise problem. Before the diffusion era, Generative Adversarial Networks (GANs) were the dominant generative model type extensively studied and utilized in real-world applications. Simultaneously, the application of Optimal Transport theory (Villani, 2008), specifically Wasserstein distance, played a pivotal role in addressing key issues in generative models. This included enhancing diversity (Arjovsky et al., 2017; Gulrajani et al., 2017), improving convergence (Sanjabi et al., 2018), and ensuring stability (Miyato et al., 2018) in GANs. To address the challenge posed by noisy datasets, several research efforts have incorporated Unbalanced Optimal Transport formulations (Chizat et al., 2018b) into the GAN, successfully reducing the impact of outliers in image synthesis (Balaji et al., 2020; Choi et al., 2023; Nietert et al., 2022).

Inspired by the Unbalanced Optimal Transport theory (Chizat et al., 2018b), we introduced RDGAN, which relies on semi-unbalanced optimal transport, as a solution to the noisy dataset problem. Our approach reformulates the adversarial training process between two consecutive diffusion timesteps based on the semi-unbalanced optimal transport formulation, thereby relaxing the strict marginal constraints. Through extensive experiments, we demonstrate that our proposed method, RDGAN, not only maintains a good Fréchet Inception Distance (FID) score (Heusel et al., 2017), a key metric for assessing image quality, but also achieves rapid training convergence compared to the standard DDGAN. Furthermore, we successfully mitigate the impact of corrupted samples present in noisy training datasets. In summary, our research has led to several noteworthy contributions:

- **Enhanced Robustness**: Firstly, we've identified a key limitation of DDGAN when confronted with noisy datasets that include outliers. To overcome this challenge, we've introduced Robust Diffusion GAN (RDGAN), a novel approach that demonstrates remarkable resilience under the presence of outliers.

- **Superior Image Generation**: Not only robust to outliers, RDGAN also generates images at higher quality compared to the baseline DDGAN on either clean or corrupted datasets, which is consistently confirmed via extensive experiments with both FID (Fréchet Inception Distance) and Recall metrics.

- **Improved Training Convergence**: Expanding upon its image generation capabilities, RDGAN also introduces substantial enhancements in training stability, surpassing its predecessor, DDGAN. This heightened stability serves to optimize the training pipeline, rendering it both more dependable and operationally efficient.

## 2 BACKGROUND

### 2.1 DIFFUSION MODELS

Diffusion models that rely on the diffusion process often take empirically thousand steps to diffuse the original data to become a neat approximation of Gaussian noise. Let's use $x_0$ to denote the true data, and $x_t$ denotes that datum after $t$ steps of rescaling data and adding Gaussian noise. The probability distributions of $x_t$ conditioned on $x_{t-1}$ and $x_0$ has the form

$$q(x_t|x_{t-1}) = \mathcal{N}(\sqrt{1 - \beta_t}x_{t-1}, \beta_t\mathbf{I}) \tag{1}$$

$$q(x_t|x_0) = \mathcal{N}(x_t; \sqrt{\overline{\alpha}_t}x_0, (1 - \overline{\alpha}_t)\mathbf{I}) \tag{2}$$

where $\alpha_t = 1 - \beta_t$, $\bar{\alpha}_t = \prod_{s=1}^t \alpha_s$, and $\beta_t \in (0, 1)$ is set to be relatively small, either through a learnable schedule or as a fixed value at each time step in the forward process. Given that the diffusion process introduces relatively minor noise with each step, we can estimate the reverse process, denoted as $q(x_{t-1}|x_t)$, by using a Gaussian process, specifically, $q(x_{t-1}|x_t, x_0)$, which in turn could

be learned through a parameterized function $p_\theta(x_{t-1}|x_t)$. Following (Ho et al., 2020), $p_\theta(x_{t-1}|x_t)$ is commonly parameterized as:

$$p_\theta(x_{t-1} \mid x_t) = \mathcal{N}(x_{t-1}; \mu_\theta(x_t, t), \sigma_t^2 \mathbf{I}), \tag{3}$$

where $\mu_\theta(x_t, t)$ and $\sigma_t^2$ represent the mean and variance of parameterized denoising model, respectively. The learning objective is to minimize the Kullback-Leibler (KL) divergence between the true denoising distribution $q(x_{t-1}|x_t)$ and the denoising distribution parameterized by $p_\theta(x_{t-1}|x_t)$.

Unlike traditional diffusion methods, DDGAN (Xiao et al., 2022) allows for larger denoising step sizes to speed up the sampling process by incorporating generative adversarial networks (GANs). DDGAN introduces a discriminator, denoted as $D_\phi$, and optimizes both the generator and discriminator in an adversarial training fashion. The objective of DDGAN can be expressed as follows:

$$\min_\phi \max_\theta \sum_{t \geq 1} \mathbb{E}_{q(\mathbf{x}_t)} \bigg\{ \mathbb{E}_{q(\mathbf{x}_{t-1}|\mathbf{x}_t)} \Big[ -\log\big(D_\phi(\mathbf{x}_{t-1}, \mathbf{x}_t, t)\big)\Big] +$$

$$\mathbb{E}_{p_\theta(\mathbf{x}_{t-1}|\mathbf{x}_t)}\Big[\log\big(D_\phi(\mathbf{x}_{t-1}, \mathbf{x}_t, t)\big)\Big]\bigg\}, \tag{4}$$

In equation 13, fake samples are generated from a conditional generator $p_\theta(x_{t-1}|x_t)$. Due to the use of large step sizes, the distribution $q(x_{t-1}|x_t)$ is no longer Gaussian. DDGAN addresses this by implicitly modeling this complex multimodal distribution using a generator $G_\theta(x_t, z, t)$, where $z$ is a $D$-dimensional latent variable drawn from a standard Gaussian distribution $\mathcal{N}(0, \mathbf{I})$. Specifically, DDGAN first generates an unperturbed sample $x_0'$ through the generator $G_\theta(x_t, z, t)$ and obtains the corresponding perturbed sample $x_{t-1}'$ using $q(x_{t-1}|x_t, x_0)$. Simultaneously, the discriminator evaluates both real pairs $D_\phi(x_{t-1}, x_t, t)$ and fake pairs $D_\phi(x_{t-1}', x_t, t)$ to guide the training process.

## 2.2 Unbalanced Optimal Transport

In this section, we provide some background on optimal transport (OT), its unbalanced formulation (UOT), and their applications.

**Optimal Transport:** Let $\mu$ and $\nu$ be two probability measures in the set of probability measures $\mathcal{P}(\mathcal{X})$ for space $\mathcal{X}$, the OT distance between $\mu$ and $\nu$ is defined as

$$\mathsf{OT}(\mu, \nu) = \min_{\pi \in \Pi(\mu, \nu)} \int c(x, y) d\pi(x, y), \tag{5}$$

where $c : \mathcal{X} \times \mathcal{X} \to [0, \infty)$ is a cost function, $\Pi(\mu, \nu)$ is the set of joint probability measures on $\mathcal{X} \times \mathcal{X}$ which has $\mu$ and $\nu$ as marginal probability measures. The dual form of OT is

$$\mathsf{OT}(\mu, \nu) = \sup_{u(x)+v(y) \leq c(x,y)} \Big[ \int_\mathcal{X} u(x) d\mu(x) + \int_\mathcal{X} v(y) d\nu(y) \Big] \tag{6}$$

Denote $v^c(y) = \inf_{y \in \mathcal{X}} \big\{ c(x, y) = v(y) \big\}$ to be the $c$-transform of $v$, then the dual formulation of OT could be written in the following form

$$\mathsf{OT}(\mu, \nu) = \sup_v \Big[ \int_\mathcal{X} v^c(x) d\mu(x) + \int_\mathcal{X} v(y) d\nu(y) \Big].$$

**Unbalanced Optimal Transport**: Another version of OT introduced by (Chizat et al., 2018b) is Unbalanced Optimal Transport (UOT) formulated as follows

$$\mathsf{UOT}(\mu, \nu) = \min_{\pi \in \mathcal{M}(\mathcal{X} \times \mathcal{X})} \int c(x, y) d\pi(x, y) + \mathsf{D}_{\Psi_1}(\pi_1 \| \mu) + \mathsf{D}_{\Psi_2}(\pi_2 \| \nu), \tag{7}$$

where $\mathcal{M}(\mathcal{X} \times \mathcal{X})$ denotes the set of joint non-negative measures on $\mathcal{X} \times \mathcal{X}$; $\pi$ is an element of $\mathcal{M}(\mathcal{X} \times \mathcal{X})$, its marginal measures corresponding to $\mu$ and $\nu$ are $\pi_1$ and $\pi_2$, respectively; the $\mathsf{D}_{\Psi_i}$ are often set as the Csiszár-divergence, i.e., Kullback-Leibler divergence, $\chi^2$ divergence. In contrast to OT, the UOT does not require hard constraints on the marginal distributions, thus allowing more flexibility to adapt to different situations. The formulation equation 7 has been applied to unbalanced measures to find developmental trajectories of cells (Schiebinger et al., 2019). Another application of UOT is robust optimal transport in cases when data are corrupted with outliers (Balaji et al., 2020)

or when mini-batch samples (K.Fatras et al., 2021; Nguyen et al., 2022) are biased representation of the data distribution. Similar to the OT, solving the UOT again could be done through its dual form (Chizat et al., 2018b; Gallouët et al., 2021; Vacher & Vialard, 2022)

$$\mathsf{UOT}(\mu, \nu) = \sup_{u(x)+v(y)\leq c(x,y)} \left[ \int_{\mathcal{X}} -\Psi_1^*(-u(x))d\mu(x) + \int_{\mathcal{X}} -\Psi_2^*(-v(y))d\nu(y) \right], \quad (8)$$

where $u, v \in \mathcal{C}(\mathcal{X})$ in which $\mathcal{C}$ denotes a set of continuous functions over its domain; $\Psi_1^*$ and $\Psi_2^*$ are the convex conjugate functions of $\Psi_1$ and $\Psi_2$, respectively. If both function $\Psi_1^*$ and $\Psi_2^*$ are non-decreasing and differentiable, we could next remove the condition $u(x) + v(y) \leq c(x, y)$ by the $c$-transform for function $v$ to obtain the semi-dual UOT form (Vacher & Vialard, 2022):

$$\mathsf{UOT}(\mu, \nu) = \sup_{v\in\mathcal{C}(\mathcal{X})} \left[ \int_{\mathcal{X}} -\Psi_1^*\big(-v^c(x)\big)d\mu(x) + \int_{\mathcal{X}} -\Psi_2^*\big(-v(y)\big)d\nu(y) \right]. \quad (9)$$

## 3 METHOD

In this section, we introduce our method, starting with Denoising Diffusion GAN (DDGAN) (Xiao et al., 2022), a hybrid approach that combines elements of GAN and Denoising Diffusion Probabilistic Models. The diffusion process employed in DDGAN helps preventing mode collapse and smoothes the data distribution, reducing the likelihood of overfitting by the discriminator. This results in improved training stability and mode coverage. However, DDGAN is susceptible to outlier samples in noisy datasets, an issue that has been illustrated through a simple toy experiment in Figure 2. The top-left subfigure in Figure 2 highlights how DDGAN, when trained on noisy datasets, inadvertently generates outlier samples. Remarkably, this challenge has remained unaddressed by prior research efforts.

### 3.1 SEMI DUAL UOT IN AVERSARIAL TRAINING

In the GAN problem, we can parametrize the generative network $G_\theta : \mathcal{X} \to \mathcal{X}$ as follows:

$$G_\theta(x) \in \arg\inf_{y\in\mathcal{X}} \big[c(x, y) - v(y)\big] \quad \Leftrightarrow \quad v^c(x) = c\big(x, G_\theta(x)\big) - v\big(G_\theta(x)\big) \quad (10)$$

and discriminative network $D_\phi = v$. Therefore, Equation 9 can be written as following:

$$\mathsf{UOT}(\mu, \nu) = \sup_{D_\phi} \left[ \int_{\mathcal{X}} \Psi_1^*\Big( -\big[c\big(x, G_\theta(x)\big) - D_\phi\big(G_\theta(x)\big)\big]\Big)d\mu(x) + \int_{\mathcal{X}} \Psi_2^*\big(-D_\phi(y)\big)d\nu(y) \right]. \quad (11)$$

$$= \inf_{D_\phi} \left[ \int_{\mathcal{X}} \Psi_1^*\Big( -\inf_{G_\theta}\big[c\big(x, G_\theta(x)\big) - D_\phi\big(G_\theta(x)\big)\big]\Big)d\mu(x) + \int_{\mathcal{X}} \Psi_2^*\big(-D_\phi(y)\big)d\nu(y) \right]. \quad (12)$$

### 3.2 ROBUST DIFFUSION GAN

DDGAN training involves matching the conditional GAN generator $p_\theta(x_{t-1}|x_t)$ and $q(x_{t-1}|x_t)$ using an adversarial loss that minimizes a divergence $D_{\mathrm{adv}}$ per denoising step.

$$\min_\theta \sum_{t\geq 1} \mathbb{E}_{q(\mathbf{x}_t)} \left[ D_{\mathrm{adv}} \left( q\left(\mathbf{x}_{t-1} \mid \mathbf{x}_t\right) \| p_\theta\left(\mathbf{x}_{t-1} \mid \mathbf{x}_t\right)\right)\right]. \quad (13)$$

$D_{\mathrm{adv}}$ can take the form of Wasserstein distance, Jensen-Shannon divergence, or an f-divergence, depending on the specific adversarial training setup. Equation 13 can be reformulated as a more general optimal transport problem:

$$\min_\theta \sum_{t\geq 1} \mathbb{E}_{q(\mathbf{x}_t)} \left[ \mathsf{OT}\left( q\left(\mathbf{x}_{t-1} \mid \mathbf{x}_t\right), p_\theta\left(\mathbf{x}_{t-1} \mid \mathbf{x}_t\right)\right)\right]. \quad (14)$$

As discussed in Section 2.2, Unbalanced Optimal Transport (UOT), introduced by (Chizat et al., 2018a), offers an alternative version of OT that doesn't necessitate rigid constraints on marginal

distributions. Consequently, it presents a viable approach for addressing noisy datasets containing outliers. Our approach, RDGAN, differs from the OT-based DDGAN approach, which requires strict marginal constraints and is susceptible to outliers. Instead, we propose to formulate the corresponding UOT training objective for DDGAN, hence the name RDGAN:

$$\min_\theta \sum_{t \geq 1} \mathbb{E}_{q(\mathbf{x}_t)} \left[ \mathsf{UOT} \left( q\left( \mathbf{x}_{t-1} \mid \mathbf{x}_t \right), p_\theta \left( \mathbf{x}_{t-1} \mid \mathbf{x}_t \right) \right) \right]. \tag{15}$$

As RDGAN uses the same network $G_\theta$ and $D_\phi$ as DDGAN, directly substitute the semi dual UOT formula 12 into the DDGAN training objective Equation 4, we obtain the Training Algorithm 1:

---

**Algorithm 1:** Training Algorithm RDGAN

---

**Input:** *The data distribution $p_{data}$. Non-decreasing, differentiable, convex function pair $(\Psi_1^*, \Psi_2^*)$. Generator network $G_\theta$ and the discriminator network $D_\phi$. Total training iteration number $K$. Batch size $B$.*

**for** $k = 0, 1, 2, \ldots, K$ **do**

    Sample $x_0 \sim p_{\text{data}}, z \sim \mathcal{N}(\mathbf{0}, \mathbf{I}_d), t \sim [1:T]$.

    Sample $x_t \sim p(\cdot|x_0), \widehat{x}_0 = G_\theta(x_t, z, t), \widehat{x}_{t-1} \sim p(\cdot|\widehat{x}_0, x_t)$.

$$\mathcal{L}_D = \frac{1}{B} \Psi_1^* \big( - c\left(x_t, \widehat{x}_0\right) + D_\phi\left(\widehat{x}_{t-1}, x_t, t\right) \big) + \frac{1}{B} \Psi_2^* \big( - D_\phi\left(x_{t-1}, x_t, t\right) \big).$$

    Update $\phi$ to minimize the loss $\mathcal{L}_D$.

$$\mathcal{L}_G = \frac{1}{B} \big( c\left(x_t, \widehat{x}_0\right) - D_\phi\left(\widehat{x}_{t-1}, x_t, t\right) \big).$$

    Update $\theta$ to minimize the loss $\mathcal{L}_G$.

**end**

---

## 4 EXPERIMENT

In this section, we first show that RDGAN not only maintains but also improves three critical generative modeling criteria: fast sampling, high-fidelity generation, and mode coverage, all while ensuring stable and fast training convergence. Then we carry out extensive experiments across various noisy dataset settings, demonstrating the heightened robustness of our RDGAN method to outliers. Finally, we conduct ablation studies, focusing on the selection of $\Psi_1^*, \Psi_2^*$ and the stable performance of RDGAN even when the ratio of outlier increases. For the experiment, we use the $\mathbf{L}_2$ distance as our cost function: $c(x, y) = \frac{1}{\tau}||x - y||_2^2$ with the hyperparameter $\tau$. Details of all experiments and evaluations can be found in Appendix A.

### 4.1 THREE KEY EVALUATION CRITERIA FOR GENERATIVE MODELS

We assessed the performance of our RDGAN technique on two distinct datasets: CIFAR-10 ($32 \times 32$) (Krizhevsky, 2012), and STL-10 ($64 \times 64$) (Coates et al., 2011) for image synthesis tasks. To gauge the effectiveness of our approach, we utilized two widely recognized metrics, namely FID (Heusel et al., 2017) and Recall (Kynkäänniemi et al., 2019). As shown in Table 1, RDGAN attains a notably lower FID score of $\mathbf{3.53}$, in contrast to the baseline method DDGAN, which registers a FID score of $3.75$. Furthermore, the recall score for RDGAN remains stable at $0.56$, closely approximating DDGAN's recall score of $0.57$. Turning our attention to the STL-10 dataset, Table 2 illustrates a substantial improvement in FID for RDGAN compared to DDGAN. Specifically, RDGAN achieves a remarkable FID score of $\mathbf{16.20}$, approximately 5 points lower than DDGAN's score of $21.79$. Additionally, our method secures a higher recall score of $0.44$, surpassing DDGAN's score of $0.40$. In summary, our proposed RDGAN method outperforms the baseline DDGAN in **high-fidelity image generation**. In Figure 3, we demonstrate that RDGAN **converges much faster** than DDGAN. By epoch $400$, RDGAN achieves an FID of less than $20$, while DDGAN's FID remains above $100$. For a visual representation of our results, please refer to Figure 1. Furthermore, the recall score of RDGAN is either better or equal to that of DDGAN, indicating that our method exhibits **robust mode coverage**. For a fair comparison, the architecture and hyperparameters used to

train RDGAN and DDGAN are identical. The sole distinction lies in our semi-dual UOT objective function, ensuring that the **fast sampling time** of DDGAN is preserved in RDGAN.

| Model | FID↓ | Recall↑ | NFE↓ |
|---|---|---|---|
| Our | **3.53** | 0.56 | 4 |
| WaveDiff (Phung et al., 2023) | 4.01 | 0.55 | 4 |
| DDGAN (Xiao et al., 2022) | 3.75 | **0.57** | 4 |
| DDPM (Ho et al., 2020) | 3.21 | 0.57 | 1000 |
| DDIM (Song et al., 2021a) | 4.67 | 0.53 | 50 |
| Recovery EBM (Gao et al., 2021) | 9.58 | - | 180 |
| StyleGAN2 (Karras et al., 2020b) | 8.32 | 0.41 | 1 |
| NVAE (Vahdat & Kautz, 2020) | 23.5 | 0.51 | 1 |

Table 1: Quantitative results on CIFAR-10 $32 \times 32$

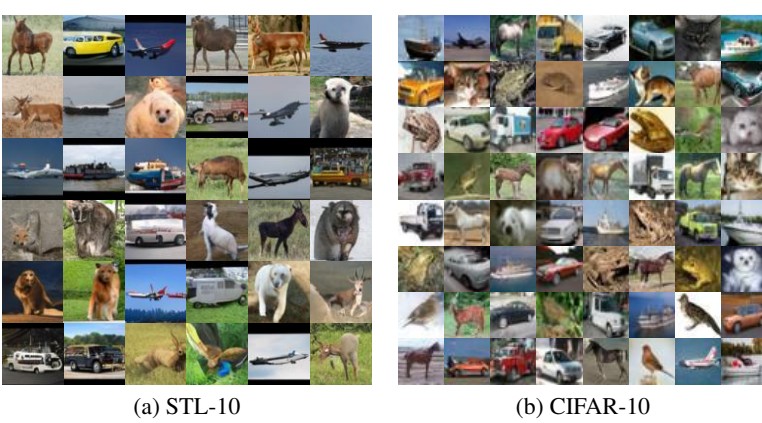

(a) STL-10          (b) CIFAR-10

Figure 1: Qualitative results of RDGAN on 2 datasets STL-10 and CIFAR-10.

| Model | FID↓ | Recall↑ |
|---|---|---|
| Our | **16.20** | **0.44** |
| DDGAN (Xiao et al., 2022) | 21.79 | 0.40 |
| TransGAN (Jiang et al., 2021) | 18.28 | - |
| SNGAN (Miyato et al., 2018) | 40.1 | - |
| StyGAN2+ADA (Karras et al., 2020a) | 13.72 | 0.36 |
| StyGAN2+Aug(Karras et al., 2020a) | 12.97 | 0.39 |

Table 2: Quantitative performance of RDGAN on STL-10 $64 \times 64$. RDGAN surpasses DDGAN at both metric FID and Recall. StyGAN is StyleGAN.

Table 3: Comparison of the training convergence on STL-10 between DDGAN and RDGAN

## 4.2 ROBUSTNESS GENERATION

To demonstrate the effectiveness of our RDGAN method on noisy datasets, we initially compare the generated density obtained by training RDGAN and DDGAN techniques with the ground truth target density on a toy dataset. As illustrated in Figure 2, we visually observe that RDGAN exclusively generates new data points that align with the clean mode on the right, whereas DDGAN produces outlier data scattered between the two modes. This experiment highlights the vulnerability of DDGAN to outliers and underscores the superior reliability of our RDGAN method in noisy training datasets.

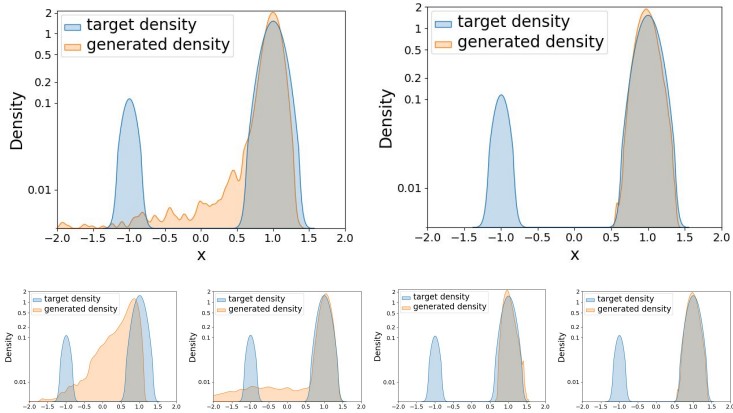

Figure 2: **Outlier Robustness on Toy Dataset** with $5\%$ outliers. The toy dataset is a mixture of two gaussians $\mathcal{N}(1, 0.1)$ (clean dataset), $\mathcal{N}(-1, 0.05)$ (outlier dataset) with the mixture rate is $(0.95, 0.05)$. In the first row, subplots compare target and generated densities between DDGAN and RDGAN. Left: DDGAN; Right: RDGAN. The second row showcases partial timestep RDGAN results. From left to right, semi dual UOT loss is applied to the first 1, 2, 3 timesteps, and then to all timesteps.

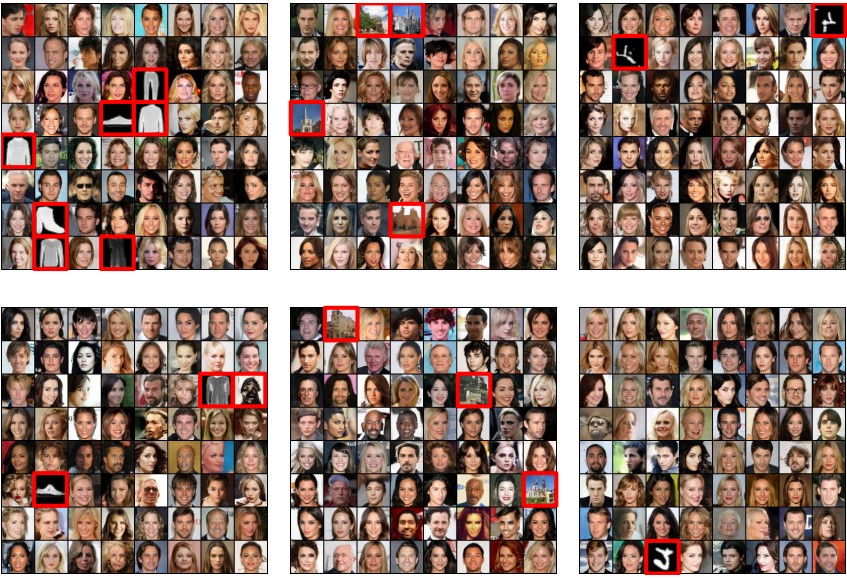

Figure 3: From left to right is corresponding to CE+FT, CE+MT and CE+CH dataset. Top: DDGAN, Bottom: RDGAN. For the dataset CI+MT, please refer to the second column Figure 4. The red boxes indicate the synthesized outliers among the clean synthesized samples.

Furthermore, we conducted experiments with RDGAN on various high-dimensional datasets perturbed with diverse noise types, mirroring real-world applications to validate its robustness in handling noisy datasets. As shown in Table 4, we trained both RDGAN and DDGAN in scenarios where there is a $5\%$ outlier presence in noisy datasets. Since the resolution of clean and outlier datasets might be different, we rescaled the clean and outlier datasets to the same resolution, with CI+MI at $32 \times 32$ resolution and the other three datasets (CE+FT, CE+MT, and CE+CH) at $64 \times 64$. The results in Table 4 demonstrate that RDGAN significantly outperforms DDGAN across all noisy datasets. For instance, on the CI+MT dataset, RDGAN surpasses DDGAN by more than 4 points in the FID score. Our technique proves effective with various outlier datasets, as evidenced by CE+FT, CE+MT, and CE+CH, where we keep the same clean dataset and change the outlier dataset. We observe that both FT and MT, despite being grayscale and visually distinct from CE, perform

|        | CI+MT | CE+FT | CE+MT | CE+CH |
|--------|-------|-------|-------|-------|
| RDGAN  | **4.42** | **7.89** | **9.29** | **7.86** |
| DDGAN  | 8.81  | 10.68 | 12.95 | 9.83  |

Table 4: We performed a FID comparison between DDGAN and RDGAN on four noisy datasets. In the table, we use abbreviated dataset names: CI (CIFAR-10 $32 \times 32$), CE (CelebHQ $64 \times 64$), MT (MNIST-10 $28 \times 28$), FT (FASHION MNIST $28 \times 28$), and CH (LSUN CHURCH $64 \times 64$). Each noisy dataset includes a $5\%$ outlier perturbation. For example, CI+MI represents CIFAR-10 with a $5\%$ MNIST outlier where the outlier images are scaled to have the same size as the clean data.

well with our method, with an FID gap of around 3 points when compared with the corresponding DDGAN model. Notably, even though the CH dataset comprises RGB images and bears great similarity to CE, our RDGAN effectively learns to automatically eliminate outliers, achieving a 2-point lower FID score than DDGAN. This demonstrates RDGAN's capability to discriminate between two datasets in the same RGB domain, which has not previously been reported by other robust generative works (Balaji et al., 2020; Le et al., 2021; Choi et al., 2023). For the qualitative result of the experiment in Table 4 , refer to the Figure 3.

## 4.3 ABLATION STUDIES

### 4.3.1 CHOICE OF $\Psi_1^*$ AND $\Psi_2^*$

Given that we set $D_{\Psi_i}$ as Csiszár-divergences, commonly used functions like KL and $\chi^2$ were tested as choices for $\Psi_1$ and $\Psi_2$ in RDGAN. However, using KL as $\Psi_i$ led to infinite loss during RDGAN training, even with meticulous hyperparameter tuning, likely due to the exponential convex conjugate form of KL (refer to Appendix B). For $\chi^2$ as $\Psi_i$, the first row of Table 5 reveals that RDGAN with $\chi^2$ achieved an FID score of $5.04$, outperforming DDGAN's FID of $8.81$ on CIFAR-10 with $5\%$ outlier MNIST. While RDGAN with $\chi^2$ excels with noisy datasets containing outliers, its performance lags behind DDGAN on clean datasets. Inspired by prior works such as (Miyato et al., 2018; Xiao et al., 2022), which employ the softplus function instead of other divergences, we conducted RDGAN experiments with $\Psi_i^* = $ Softplus. Interestingly, this approach proved highly effective, surpassing both DDGAN and RDGAN with $\chi^2$ on both clean and noisy datasets.

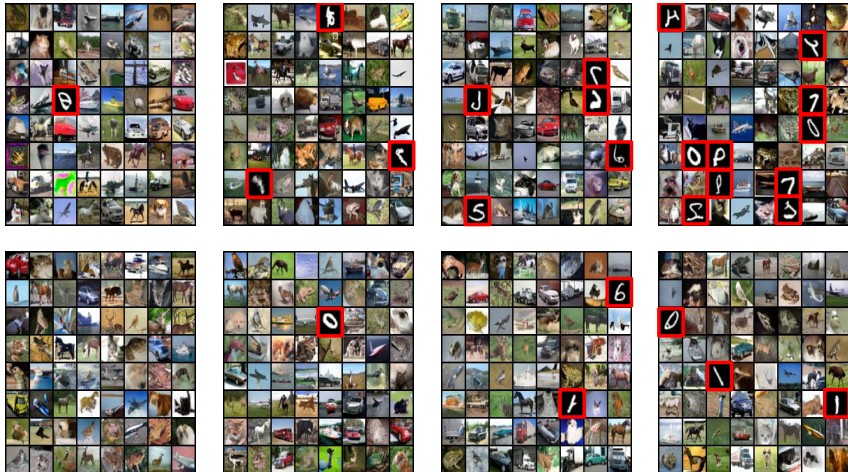

Figure 4: From left to right is corresponding to $3\%, 5\%, 7\%$ and $10\%$ MNIST outlier in noisy CIFAR-10 dataset. The first row is the DDGAN and the second row is the RDGAN. The red boxes indicate the synthesized outliers among the clean synthesized samples.

| $\Psi_1^*$ | $\Psi_2^*$ | FID (clean) ↓ | FID (5%) ↓ |
|---|---|---|---|
| $\chi^2$ | $\chi^2$ | 3.93 | 5.04 |
| softplus | softplus | **3.53** | **4.82** |
| None | None | 3.75 | 8.81 |

Table 5: The choice of $\Psi_1^*$ and $\Psi_2^*$. The FID (clean) and FID (5%) are computed FID on clean CIFAR-10 and CIFAR10 with 5% MNIST outlier. The last row in the table is the DDGAN

| | Outlier Ratio | | FID | |
|---|---|---|---|---|
| Perturbed | DDGAN | RDGAN | DDGAN | RDGAN |
| 3% | 3.2% | 0.2% | 4.76 | 3.89 |
| 5% | 4.1% | 1.7% | 8.81 | 4.82 |
| 7% | 6.9% | 2.3% | 9.55 | 5.17 |
| 10% | 9.8% | 3.8% | 14.77 | 6.09 |

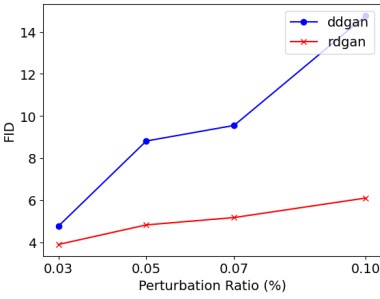

Table 6: Synthesized Outlier Ratios (Left) and FID Scores (Right) for DDGAN and RDGAN. Rows indicate outlier percentages in noisy training data ($3\%, 5\%, 7\%$, and $10\%$) from top to bottom.

Figure 5: We plot the FID comparison between RDGAN and DDGAN on CIFAR-10 data perturbed by MNIST.

### 4.3.2 Perturbation Ratio

As shown in Table 6 and Figure 5, RDGAN consistently maintains strong performance even as the outlier percentage in noisy datasets increases. While the perturbed ratio in the training dataset escalates from 3% to 10%, RDGAN's FID only increases by around 2 points. In contrast, DDGAN's FID increases by more than 10 points, and the synthesized outlier ratio of RDGAN rises from 0.2% to 3.8% compared to DDGAN's increase from 3.2% to 9.8%.

### 4.3.3 Partial Timestep RDGAN

As DDGAN comprises multiple diffusion steps which are trained with adversarial networks, and RDGAN introduces the semi-dual UOT loss in place of the traditional GAN loss, a natural question arises: How well does RDGAN perform when only partially applying the proposed loss within the DDGAN framework? Referring to Figure 2, it becomes evident that the performance of RDGAN with partial timesteps falls behind that of RDGAN with all timesteps, referred to simply as **"RDGAN"** in other sections of this paper. This discrepancy may be attributed to the fact that RDGAN has demonstrated the ability to either maintain or surpass DDGAN performance on both clean and perturbed images, making full-timestep RDGAN the superior choice over partial-timestep RDGAN.

## 5 Conclusion

In this paper, we have highlighted the limitations of the DDGAN model when faced with noisy datasets containing outliers. To address this issue, we have introduced the RDGAN technique, which incorporates Semi Dual Unbalanced Optimal Transport into the DDGAN framework. RDGAN has demonstrated the ability to either maintain or enhance performance across all three critical generative modeling criteria: mode coverage, high-fidelity generation, and fast sampling, all while ensuring rapid training convergence. Additionally, our paper showcases that RDGAN significantly outperforms DDGAN on noisy training datasets with various settings, making it a promising approach for real-world noisy datasets. Moreover, our work is complementary to other DDGAN improvement techniques, such as WaveDiff (Phung et al., 2023), suggesting potential for future investigations into combining these approaches to create an even more robust generative framework.

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
