# OpenReview forum: "ROBUST DIFFUSION GAN USING SEMI-UNBALANCED OPTIMAL TRANSPORT"
_ICLR.cc/2024/Conference — ICLR 2024 Conference Withdrawn Submission_

### Official Review · Reviewer_gFqX · 2023-10-23

**Soundness:** 3 good
**Presentation:** 2 fair
**Contribution:** 2 fair
**Rating:** 5
**Confidence:** 4

**Summary:**

This article introduces a robust training technique based on semi-unbalanced optimal transport to mitigate the impact of outliers effectively.
Meanwhile DDGAN remains susceptible to performance drops caused by datasets that are corrupted with outlier samples.
Through comprehensive evaluations, the RDGAN demonstrate that it outperforms vanilla DDGAN in terms of the aforementioned
generative modeling criteria, i.e., image quality, mode coverage of distribution, and inference speed, and exhibits improved robustness when dealing with both clean and corrupted datasets.

**Strengths:**

Given the recent advancements in generative models such as Dalle2, stable diffusion, or diffusion GANs,  has effectively
addressed the challenges of generative modeling, namely producing high-quality
samples, covering different data modes. But it remains susceptible to performance drops caused by datasets that are corrupted
with outlier samples.
This article introduces a novel approach by employing a semi-unbalanced optimal transport to mitigate the impact of outliers effectively.

**Weaknesses:**

The approach employed in the paper may appear to lack novelty. This is because the RDGAN simply replaces the loss function of diffusion GANs with a semi-unbalanced optimal transport, without conducting a thorough analysis of the relationship between the semi-unbalanced optimal transport and diffusion GANs from both empirical and theoretical perspectives.

**Questions:**

1. I would suggest that the author conducting a thorough analysis of the relationship between the semi-unbalanced optimal transport and diffusion GANs from both empirical and theoretical perspectives in the corrupted with outlier samples.

2. In Tables 1 and 2, RDGAN still does not achieve the best results when compared to other methods.

3. I recommend that the author present a comparison of results for different diffusion step values (T) in DRGAN, DDGAN, and the ablation study.

4. I recommend that the author provide a explanation of why the semi-dual UOT objective
ensuring that the fast sampling time of DDGAN is preserved in RDGAN?

5. "In contrast, DDGAN’s
FID increases by more than 10 points, and the synthesized outlier ratio of RDGAN rises from 0.2
to 3.8 compared to DDGAN’s increase from 3.2 to 9.8."
Does this mean that DDGAN is more likely to synthesize more samples in outlier data compared to RDGAN? Is this considered a desirable capability, or is it potentially problematic?

6. Can you present the outcomes achieved when training StyGAN2+Aug (Karras et al., 2020a) on mixed datasets, and include them in Table 4? Since stylegan2+AUG appears to generate a greater diversity of samples in Table 2, it would be valuable to assess its performance on mixed datasets as well.

---

### Official Review · Reviewer_P1f3 · 2023-10-26

**Soundness:** 3 good
**Presentation:** 2 fair
**Contribution:** 2 fair
**Rating:** 5
**Confidence:** 4

**Summary:**

This paper proposed a new method that combine UOT with diffusion GANs in order to permit more robust training while achieving high perception quality as well as fast inference.  The method was empirically evaluated on image generation.

**Strengths:**

The strength of this paper is its technical soundness. The motivation is properly justified. The proposed method seems to make sense as a solution to the robustness problem to be solved.

**Weaknesses:**

The main weakness of this paper is the novelty of the proposed method which seems to be a direct marriage of two existing ideas.  Indeed, this hasn't been done. However, I'm not sure how much this field of research will benefit from this obvious extension, especially when the practical side of this paper is also weak.  The experiments were done in very low dimensional settings, which is fine if the novelty of the approach is great.  When both are lacking, I'm reluctant to accept it to be published with the current version.

**Minor**:
The writing of the paper can be improved, including the organisation of the paper flow, as well as the typos such as a wrongly referenced equation ("equation 13" in the paragraph before section 2.2),  the c-transform of v (following equation 6), and introducing "Unbalanced Optimal Transport (UOT)" twice.

Some languages used need to be a bit more rigorous. For instance, in the paragraph 2 in the introduction - "slow computational speed" is confusing. The computational speed for training a diffusion model isn't slow in comparison.  It's only the inference/sampling speed that's the problem.  Second example is when you say "Equation 13 can be reformulated as a **more** general optimal transport problem:".  Obviously equation 13 is more general as D_{adv} can be one of the many distances/divergences.

**Questions:**

Can you comment on the performance of StyGAN2+ADA and StyGAN2+Aug in Table 2, in comparison to that of RDGAN and of DDGAN?  And how do you think about the disagreement in FID and Recall?

---

### Official Review · Reviewer_ov4k · 2023-11-01

**Soundness:** 2 fair
**Presentation:** 2 fair
**Contribution:** 2 fair
**Rating:** 3
**Confidence:** 3

**Summary:**

This paper introduces a robust training technique based on semi-unbalanced optimal transport to mitigate the impact of outliers. Through comprehensive evaluations, this paper demonstrates that the proposed RDGAN outperforms vanilla DDGAN in terms of the FID and recall, meanwhile being robust to outliers.

**Strengths:**

The robustness of diffusion generative models is an important topic but is relatively less studied in the literature. This paper presents a simple modification to the existing DDGAN method, by using the semi-unbalanced optimal transport, to improve the method's robustness to outliers.

Empirically, a suite of numerical results is presented to show the robust performance.

**Weaknesses:**

The contribution of this paper is limited. This paper replaces the reverse KL divergence in vanilla DDGAN with the unbalanced optimal transport. However, there are not much insights stated in the paper for using UOT.

Moreover, the experiments seem to be insufficient as well, there is a lack of comparison with the type of methods such as Wasserstein GAN, OT-GAN (and UOT-GAN if possible). In addition, in terms of the robustness of the proposed method and the ablation studies, this paper only compares with the vanilla DDGAN method, which is a bit limited.

**Questions:**

Please see the above in the weakness section. My main questions are: (1) is there any insight for using UOT objective within the DDGAN framework, why would it improve the overall image quality and convergence speed (even under the scenarios without outliers), and (2) is it possible to also compare with OT-GAN type methods since they also use OT type divergence for discriminators.

---

### Official Review · Reviewer_MvTD · 2023-11-05

**Soundness:** 3 good
**Presentation:** 2 fair
**Contribution:** 2 fair
**Rating:** 5
**Confidence:** 4

**Summary:**

The paper proposes to replace the optimal transport formulation in DDGAN with Unbalanced optimal transport formulation.

**Strengths:**

1. The paper proposes a way to address the noisy dataset generative problem with the unbalanced optimal transport.

2. The paper is well-organized and well written.

**Weaknesses:**

1. The novelty. The method replaces the extsing optimal transport loss with the unbalanced optimal transport.  The technical novelty is limited.  Or authors may consider adding more in-depth analysis about the unbalanced optimal transport in diffusion model. section 4.3.1 is a good example.

2. The noisy datasets are synthetic. Authors combines digits and CIFAR dataset, which are usually unlikely to happen in the real world. It would be better if authors could add experiments on some more real-world noisy datasets.

3. Some noisy-learning baselines need to be included. For instance, can we apply some noisy sample detection (a simplest way would be clustering and I think it should be easy to cluster the cifar images and digit image into two different groups.) before learning the datasets instead of using unbalanced optimal transport?

Minor:

1. It would be better to introduce motivation to learning the generative model under noisy samples.

**Questions:**

as above